# PCPs: Patient Cardiac Prototypes

## Abstract

Many clinical deep learning algorithms are population-based and difficult to interpret. Such properties limit their clinical utility as population-based findings may not generalize to individual patients and physicians are reluctant to incorporate opaque models into their clinical workflow. To overcome these obstacles, we propose to learn patient-specific embeddings, entitled patient cardiac prototypes (PCPs), that efficiently summarize the cardiac state of the patient. To do so, we attract representations of multiple cardiac signals from the same patient to the corresponding PCP via supervised contrastive learning. We show that the utility of PCPs is multifold. First, they allow for the discovery of similar patients both within and across datasets. Second, such similarity can be leveraged in conjunction with a hypernetwork to generate patient-specific parameters, and in turn, patient-specific diagnoses. Third, we find that PCPs act as a compact substitute for the original dataset, allowing for dataset distillation.

## 1 Introduction

Modern medical research is arguably anchored around the "gold standard" of evidence provided by randomized control trials (RCTs) (Cartwright, 2007). However, RCT-derived conclusions are population-based and fail to capture nuances at the individual patient level (Akobeng, 2005). This is primarily due to the complex mosaic that characterizes a patient from demographics, to physiological state, and treatment outcomes. Similarly, despite the success of deep learning algorithms in automating clinical diagnoses (Galloway et al., 2019; Attia et al., 2019a;b; Ko et al., 2020), network-generated predictions remain population-based and difficult to interpret. Such properties are a consequence of a network's failure to incorporate patient-specific structure during training or inference. As a result, physicians are reluctant to integrate such systems into their clinical workflow. In contrast to such reluctance, personalized medicine, the ability to deliver the right treatment to the right patient at the right time, is increasingly viewed as a critical component of medical diagnosis (Hamburg & Collins, 2010).

The medical diagnosis of cardiac signals such as the electrocardiogram (ECG) is of utmost importance in a clinical setting (Strouse et al., 1939). For example, such signals, which convey information about potential abnormalities in a patent's heart, also known as cardiac arrhythmias, are used to guide medical treatment both within and beyond the cardiovascular department (Carter, 1950). In this paper, we conceptually borrow insight from the field of personalized medicine in order to learn patient representations which allow for a high level of network interpretability. Such representations have several potential clinical applications. First, they allow clinicians to quantify the similarity of patients. By doing so, network-generated predictions for a pair of patients can be traced back to this similarity, and in turn, their corresponding ECG recordings. Allowing for this inspection of ECG recordings aligns well with the existing clinical workflow. An additional application of patient similarity is the exploration of previously unidentified patient relationships, those which may lead to the discovery of novel patient sub-cohorts. Such discoveries can lend insight into particular diseases and appropriate medical treatments. In contrast to existing patient representation learning methods (Zhu et al., 2016; Suo et al., 2017), we concurrently optimize for a predictive task (cardiac arrhythmia classification), leverage patient similarity, and design a system specifically for 12-lead ECG signals.

**Contributions.** Our contributions are the following:

1. **Patient cardiac prototypes (PCPs)** - we learn representations that efficiently summarize the cardiac state of a patient in an end-to-end manner via contrastive learning.

2. **Patient similarity quantification** - we show that, by measuring the Euclidean distance between PCPs and representations, we can identify similar patients across different datasets.

3. **Dataset distillation** - we show that PCPs can be used to train a network, in lieu of the original dataset, and maintain strong generalization performance.

## 2 RELATED WORK

**Contrastive learning** is a self-supervised method that encourages representations of instances with commonalities to be similar to one another. This is performed for each instance and its perturbed counterpart (Oord et al., 2018; Chen et al., 2020a;b; Grill et al., 2020) and for different visual modalities (views) of the same instance (Tian et al., 2019). Such approaches are overly-reliant on the choice of perturbations and necessitate a large number of comparisons. Instead, Caron et al. (2020) propose to learn cluster prototypes. Most similar to our work is that of Cheng et al. (2020) and CLOCS (Kiyasseh et al., 2020) which both show the benefit of encouraging patient-specific representations to be similar to one another. Although DROPS (Anonymous, 2021) leverages contrastive learning, it does so at the patient-attribute level. In contrast to existing methods, we learn patient-specific representations, PCPs, in an end-to-end manner

**Meta-learning** designs learning paradigms that allow for the fast adaptation of networks. Prototypical Networks (Snell et al., 2017) average representations to obtain class-specific prototypes. During inference, the similarity of representations to these prototypes determines the classification. Relational Networks (Sung et al., 2018) build on this idea by learning the similarity of representations to prototypes through a parametric function. Gidaris & Komodakis (2018) and Qiao et al. (2018) exploit hypernetworks (Ha et al., 2016) and propose to generate the parameters of the final linear layer of a network for few-shot learning on visual tasks. In contrast, during inference only, we compute the cosine similarity between representations and PCPs and use the latter as the input to a hypernetwork.

**Patient similarity** aims at discovering relationships between patient data (Sharafoddini et al., 2017). To quantify these relationships, Pai & Bader (2018) and (Pai et al., 2019) propose Patient Similarity Networks for cancer survival classification. Exploiting electronic health record data, Zhu et al. (2016) use Word2Vec to learn patient representations, and Suo et al. (2017) propose to exploit patient similarity to guide the re-training of models, an approach which is computationally expensive. Instead, our work naturally learns PCPs as efficient descriptors of the cardiac state of a patient.

## 3 METHODS

### 3.1 LEARNING PATIENT CARDIAC PROTOTYPES VIA CONTRASTIVE LEARNING

We assume the presence of a dataset, $\mathcal{D} = \{x_i, y_i\}_{i=1}^N$, comprising $N$ ECG recordings, $x$, and cardiac arrhythmia labels, $y$, for a total of $P_{\text{tot}}$ patients. Typically, multiple recordings are associated with a single patient, $p$. This could be due to multiple recordings within the same hospital visit or multiple visits to a hospital. Therefore, each patient is associated with $N/P_{\text{tot}}$ recordings. We learn a feature extractor $f_\theta : x \in \mathbb{R}^D \to h \in \mathbb{R}^E$, parameterized by $\theta$, that maps a $D$-dimensional recording, $x$, to an $E$-dimensional representation, $h$. In the quest to learn patient-specific representations, we associate each patient, $p$, out of a total of $P$ patients in the *training* set with a unique and learnable embedding, $v \in \mathbb{R}^E$, in a set of embeddings, $V$, where $|V| = P \ll N$. Such embeddings are designed to be efficient descriptors of the cardiac state of a patient, and we thus refer to them as patient cardiac prototypes or PCPs.

We propose to learn PCPs in an end-to-end manner via contrastive learning. More specifically, given an instance, $x_i$, that belongs to a particular patient, $k$, we encourage its representation, $h_i = f_\theta(x_i)$, to be similar to the same patient's PCP, $v_k$, and dissimilar to the remaining PCPs, $v_j$, $j \neq k$. We quantify this similarity, $s(h_i, v_k)$, by using the cosine similarity with a temperature parameter, $\tau$. The intuition is that each PCP, in being attracted to a diverse set of representations that belong to the same patient, should become invariant to insidious intra-patient differences. For a mini-batch of size, $B$, the contrastive loss is as follows.

$$\mathcal{L}_{contrastive} = -\sum_i^B \log \left[ \frac{e^{s(h_i, v_k)}}{\sum_j^P e^{s(h_i, v_j)}} \right] \quad (1) \qquad s(h_i, v_j) = \frac{f_\theta(x_i) \cdot v_j}{\|f_\theta(x_i)\|\|v_j\|} \cdot \frac{1}{\tau} \quad (2)$$

## 3.2 Generating Patient-specific Parameters via Hypernetworks

Network parameters are typically updated during training and fixed during inference. This allows the parameters to exploit population-based information in order to learn high-level features useful for solving the task at hand. Such an approach, however, means that all instances are exposed to the same set of parameters during inference, regardless of instance-specific information. Such information can be related to any meta-label including, but not limited to, patient ID, geographical location, and even temporal period. As an exemplar, and motivated by the desire to generate patient-specific diagnoses, we focus on patient-specific information. We are essentially converting a traditional classification task to one that is conditioned on patient-specific information. To perform such conditioning, we propose to exploit both PCPs and hypernetworks, as explained next.

We assume the presence of a hypernetwork, $g_\phi : h \in \mathbb{R}^E \to \omega \in \mathbb{R}^{E \times C}$, parameterized by $\phi$, that maps an $E$-dimensional representation, $h$, to a matrix of classification parameters, $\omega$, where $C$ is the number of class labels. During training, we feed a representation, $h_i$, to the hypernetwork and generate instance-specific parameters, $\omega_i$ (see Fig. 1 left). During inference, however, we retrieve, and feed into the hypernetwork, the most similar PCP, $v_k$, to the current representation, $h_i$, (based on similarity metric, $s$). We chose this strategy after having experimented with several of them (see Sec. 5.2). It is worthwhile to note that although this approach bears some resemblance to clustering, it is distinct from it. In a clustering scenario, we would have assigned labels to instances based on their proximity to PCPs. In contrast, we are leveraging this proximity to determine the input of a hypernetwork (see Fig. 1 right).

$$\omega_i = \begin{cases} g_\phi(h_i) & \text{for training} \\ g_\phi(v_k) & \text{for inference, } v_k = \arg\max_{v_j} s(h_i, v_j) \end{cases} \quad (3)$$

By performing this retrieval, we exploit the similarity between patients in the training and inference set. As a result, the hypernetwork generates *patient-specific* parameters that parameterize the linear classifier, $p_\omega : h \in \mathbb{R}^E \to y \in \mathbb{R}^C$, which maps a representation, $h$, to a posterior class distribution, $y$. We train the entire network in an end-to-end manner using a combined contrastive and supervised loss.

$$\mathcal{L}_{supervised} = -\sum_i^B \log p_{\omega_i}(y_i = c | h_i) \quad (4) \qquad \mathcal{L}_{combined} = \mathcal{L}_{contrastive} + \mathcal{L}_{supervised} \quad (5)$$

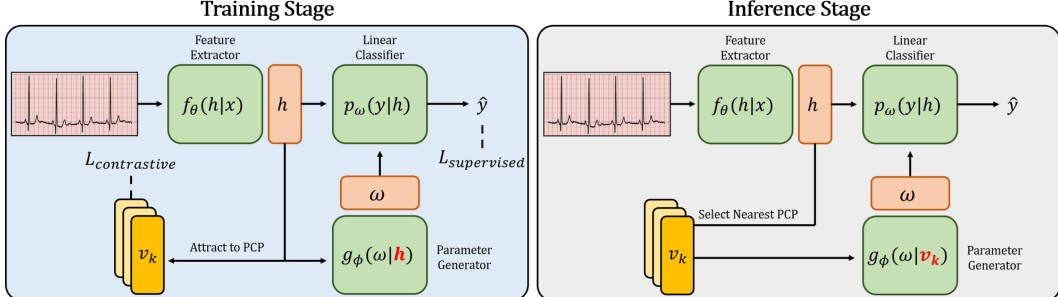

Figure 1: Training and inference stages of the personalized diagnosis pipeline. During training, instance representations, $h$, have a dual role. They are fed into a hypernetwork, $g_\phi$, to generate parameters for a linear classification layer, $p_\omega$, that outputs a prediction, $\hat{y}$. They are also encouraged to be similar to their corresponding patient cardiac prototype (PCP), $v$. During inference, the nearest PCP, $v_k$, to the representation is fed into the hypernetwork, thus generating patient-specific parameters for classification.

## 4 EXPERIMENTAL DESIGN

### 4.1 DATASETS

We conduct experiments using PyTorch (Paszke et al., 2019) on three large-scale ECG datasets that contain a significant number of patients. **PhysioNet 2020 ECG** consists of 12-lead ECG recordings from 6,877 patients alongside labels corresponding to 9 different classes of cardiac arrhythmia. Each recording can be associated with multiple labels. **Chapman ECG** (Zheng et al., 2020) consists of 12-lead ECG recordings from 10,646 patients alongside labels corresponding to 11 different classes of cardiac arrhythmia. As is suggested by Zheng et al. (2020), we group these labels into 4 major classes. **PTB-XL ECG** (Wagner et al., 2020) consists of 12-lead ECG recordings from 18,885 patients alongside 71 different types of annotations provided by two cardiologists. We follow the training and evaluation protocol presented by Strodthoff et al. (2020) where we leverage the 5 diagnostic class labels. We alter the original setup to only consider ECG segments with one label assigned to them and convert the task into a binary classification problem. Further details can be found in Appendix A.1.

Unless otherwise mentioned, datasets were split into training, validation, and test sets according to patient ID using a 60, 20, 20 configuration. In other words, patients appeared in only one of the sets. Further details about the dataset splits can be found in Appendix A.2.

### 4.2 HYPERPARAMETERS

When calculating the contrastive loss, we chose $\tau = 0.1$ as in Kiyasseh et al. (2020). We also use the same neural network architecture for all experiments. Further implementation details can be found in Appendix B.

## 5 EXPERIMENTAL RESULTS

### 5.1 PATIENT CARDIAC PROTOTYPES ARE DISCRIMINATIVE

During training, we optimize an objective function that consists of both a supervised and contrastive loss term (see Eq. 5). Based on the former, we expect representations to exhibit discriminative behaviour for the task at hand. The latter term encourages these representations to be similar to PCPs, and thus we also expect PCPs to be discriminative. In Fig. 2, we illustrate the representations of instances in the training set and the PCPs after being projected to a 2-dimensional subspace using t-SNE and colour-coded according to their class label. We find that both training representations, $h$, and PCPs, $v$, are class-discriminative. This can be seen by the high separability of the projected representations along class boundaries. Based on this finding alone, one could make the argument that PCPs are trivially detecting class label differences between patients.

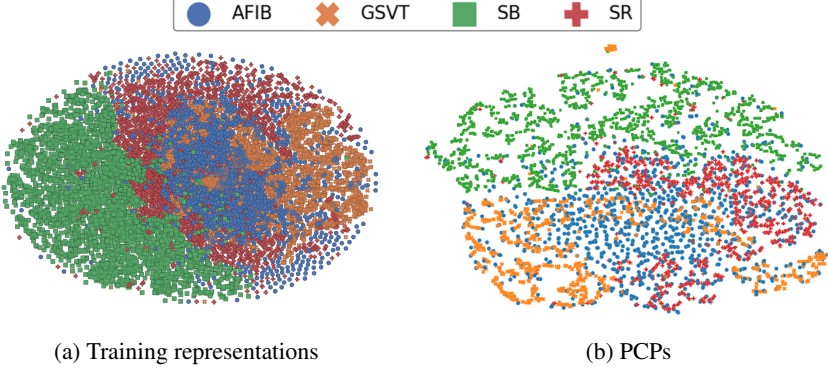

(a) Training representations        (b) PCPs

Figure 2: t-SNE projection of (a) representations, $h \in \mathbb{R}^{128}$, of instances in the training set of the Chapman dataset and (b) PCPs, $v \in \mathbb{R}^{128}$, learned on the training set, colour-coded according to the arrhythmia label assigned to each patient. Learned PCPs are also class-discriminative.

## 5.2 EFFECT OF HYPERNETWORK INPUT STRATEGY ON PERFORMANCE

As described, our pipeline uses the PCP nearest to each representation as input to the hypernetwork. This approach places a substantial dependency on that single chosen PCP. Therefore, we explore three additional input strategies that incorporate PCPs differently. **Nearest 10** searches for, and takes the average of, the 10 PCPs that are nearest to the representation. **Mean** simply takes the average of all PCPs. **Similarity-Weighted Mean** takes a linear combination of all PCPs, weighted according to their cosine similarity to the representation. In Fig. 3, we show the effect of these strategies on the test set AUC as the embedding dimension, $E$, is changed.

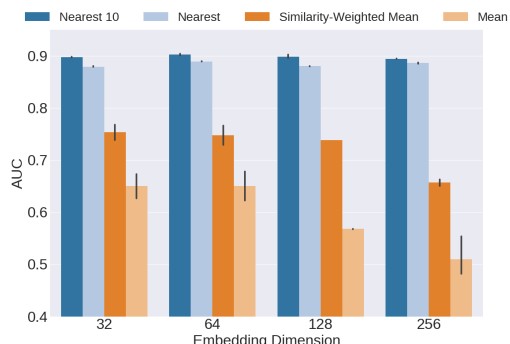

We find that exploiting the similarity of representations during inference benefits the generalization performance of the network. This is shown by the inferiority of the mean strategy relative to the remaining strategies. For example, at $E = 256$, the Mean strategy achieves an AUC $\approx 0.50$, equivalent to a random guess. However, simply weighting those PCPs according to their similarity to representations, as exemplified by Similarity Weighted Mean achieves an AUC $\approx 0.65$. This implies that representations are capturing patient-specific information.

We also find that it is more advantageous to exploit similarity to identify the nearest PCPs than to weight many PCPs. In Fig. 3, the Nearest and Nearest 10 input strategies perform best, with the latter achieving an AUC $\approx 0.90$, regardless of the embedding dimension. We hypothesize that such behaviour can be attributed to the notion that fewer PCPs are less likely to overwhelm the hy-

Figure 3: AUC on test set of Chapman dataset as a function of hypernetwork input strategies and embedding dimension, $E$. Bars are averaged across five seeds and the error bars illustrate one standard deviation. The Nearest 10 input strategy outperforms its counterparts and is unaffected by changes to the embedding dimension.

pernetwork. This, in turn, allows the hypernetwork to generate reasonable parameters for the linear classification layer. Moreover, the strong performance of these strategies despite their high dependence on so few PCPs reaffirms the utility of the learned representations.

## 5.3 PATIENT CARDIAC PROTOTYPES ARE PATIENT-SPECIFIC

So far, we have shown that PCPs are class-discriminative and can assist in achieving strong generalization performance. In this section, we aim to validate our initial claim that PCPs are patient-specific. To do so, we calculate the Euclidean distance between each PCP and two sets of representations. The first includes representations corresponding to the same patient as that of the PCP (**PCP to Same Training Patient**). The second includes representations that correspond to all remaining patients (**PCP to Different Training Patients**). In Fig. 4, we illustrate the distribution of these distances.

We find that PCPs are indeed patient-specific. This can be seen by the smaller average distance between PCPs and representations of the same patient ($\approx 4.5$) than between PCPs and representations of different patients ($\approx 9.5$). Such a finding implies that PCPs are, on average, a factor of two more similar to representations from the same patient than they are to those from other patients.

We also set out to investigate whether computing their similarity to representations of instances in the validation set (as is done in Fig. 1) was appropriate. In Fig. 4, we overlay the distribution of distances between the PCPs and representations of instances from the validation set (**PCP to Validation Patients**). We find that comparing PCPs to representations of instances in the validation set is appropriate. This is emphasized by how the average Euclidean distance between these two entities ($\approx 9$) is on the same order of the average Euclidean distance between PCPs and representations of instances in the *training* set ($\approx 4$). Based on the distributions in Fig. 4, we can also confirm that patients in the validation set are not present in the training set, as was enforced by design. This can be seen by the minimal overlap between the blue and purple distributions. Such a finding suggests that PCPs can be deployed to detect out-of-distribution patients or distribution shift. We also illustrate the generalizability of these findings by arriving at the same conclusion on the PTB-XL and PhysioNet2020 datasets (see Appendix C).

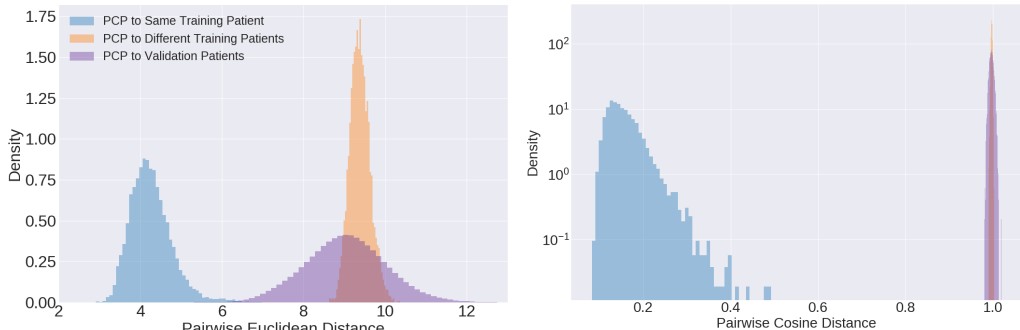

Figure 4: Distribution of pairwise (left) Euclidean and (right) cosine distance from the learned PCPs on the Chapman dataset to three sets of representations: those in the training set that belong to the same patient (blue), those in the training set that belong to different patients (orange), and those in the validation set (purple). PCPs are patient-specific since they are closer to representations belonging to the same patient than they are to representations belonging to different patients.

## 5.4 DISCOVERY OF SIMILAR PATIENTS VIA PATIENT CARDIAC PROTOTYPES

Having established that PCPs are patient-specific and class-discriminative, we now investigate whether they can be exploited to quantify patient similarity. From a clinical perspective, such information can allow physicians to discover similar patient sub-cohorts and guide medical treatment. This is particularly consequential when patients are located across different healthcare institutions. Patient similarity quantification also has the potential to add a layer of interpretability to exigent network-generated diagnoses. In this section, we exploit, and validate the ability of, PCPs to identify similar (and dissimilar) patients.

To quantify patient similarity, we compute the pairwise distance (e.g., Euclidean) between each PCP and each representation in the validation set. The distribution of these distances can be found in Fig. 5 (top). We average these distances across representations that belong to the same patient, and generate a matrix of distances between pairs of patients (see Fig. 5 centre for a subset of that matrix). Validating our findings, however, is non-trivial since similarity can be based on patient demographics, physiology, or treatment outcomes. With this in mind, we decide to validate our findings both qualitatively and quantitatively. For the former, we locate the cell in the distance matrix with the lowest distance, and in turn, identify the most similar pair of patients. We then visualize their corresponding 12-lead ECG recordings (Fig. 5 bottom).

We find that PCPs are able to sufficiently distinguish between unseen patients and thus act as reasonable patient-similarity tools. In Fig. 5 (centre), we see that there exists a large range of distance values for any chosen PCP (row). In other words, it is closer to some representations than to others, implying that a chosen PCP is not trivially equidistant to all other representations. However, distinguishing between patients is not sufficient for a patient-similarity tool. We show that PCPs can also correctly capture the relative similarity to these patients. In Fig. 5 (bottom), we show that the two patients identified as being most similar to one another, using our method, have ECG recordings with a similar morphology and arrhythmia label, supra-ventricular tachycardia. We hypothesize that this behaviour arises due to the ability of PCPs to efficiently summarize the cardiac state of a patient. Such a finding reaffirms the potential of PCPs as patient-similarity tools. We also repeat the above procedure in attempt to discover similar and dissimilar patients across *different* datasets. In doing so, we arrive at positive results and similar conclusions to those above (see Appendix D).

We now move on to quantitatively validate the PCP-derived patient similarity values. Conceptually, we build on our qualitative findings and assume that a pair of patients, identified as being similar by our method, are in fact similar if they happen to be associated with the same cardiac arrhythmia label. More specifically, we retrieve all pairs of patients that are more similar than some threshold distance, $d_E$, and determine what percentage of such retrieved pairs consist of patients with a matching cardiac arrhythmia label (Precision). In Fig. 6, we illustrate this precision as a function of different threshold distances.

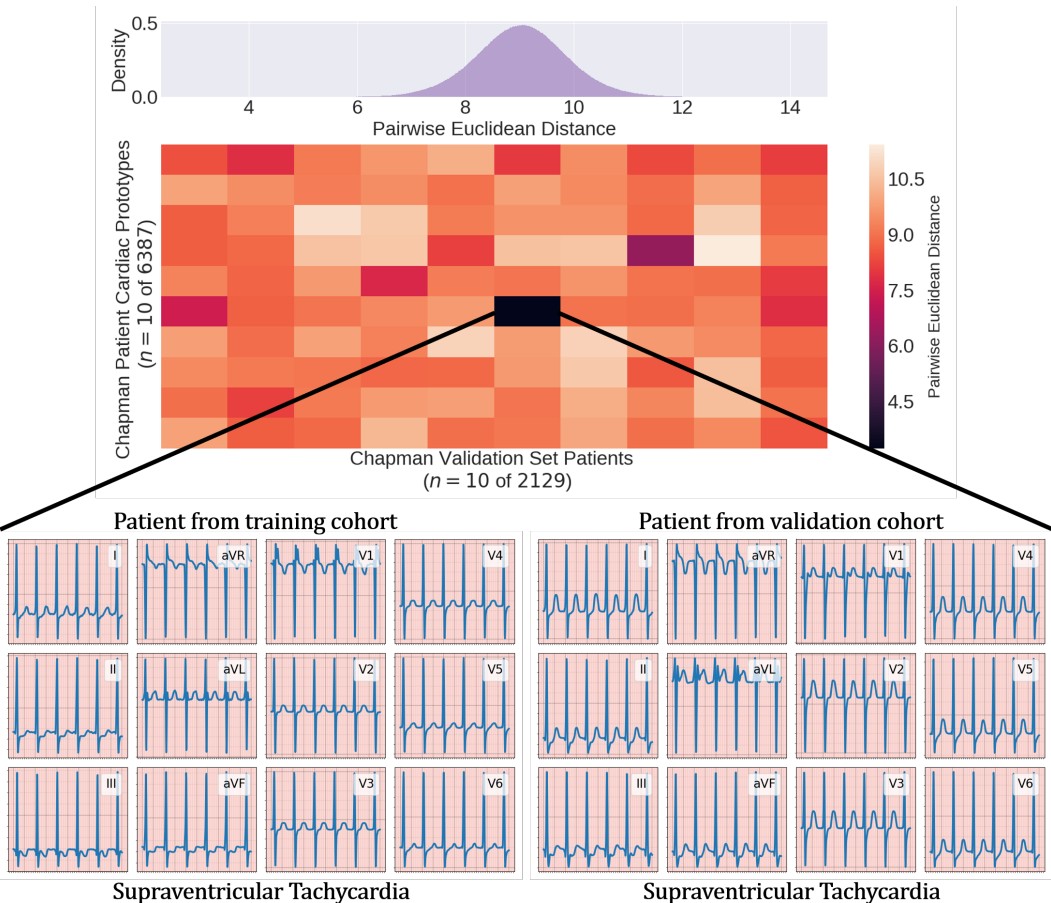

Figure 5: (**Top**) Distribution of all pairwise Euclidean distances between PCPs and representations in the validation set. (**Centre**) Matrix illustrating average pairwise distances between a subset of PCPs and representations of patients in the validation set. (**Bottom**) Visualization of the 12-lead ECG recordings of the two patients identified as being most similar by our method. Both recordings are similar and correspond to the same arrhythmia, supra-ventricular tachycardia, thus lending support to PCPs as a reliable patient-similarity tool.

We find that PCP-derived similarity values are able to identify patients with matching cardiac arrhythmia labels. For example, $> 90\%$ of the pairs of patients that are deemed very similar to one another (i.e., $d_E < 6.0$) exhibit a perfect cardiac arrhythmia label match. As we increase the threshold distance, $d_E \to 8.5$, we see that $\text{Precision} \to 0$. Such a decay is expected of a reasonable similarity metric where patients that are deemed dissimilar do not match according to their cardiac arrhythmia labels. Moreover, based on an acceptable level of precision, (e.g., $0.90$), we can identify an appropriate threshold distance (e.g., $d_E \approx 6.2$). It is worthwhile to note that this specific threshold coincides with the region of minimal distribution overlap we observed in Fig. 4. This suggests that a simple threshold can be derived from those distributions.

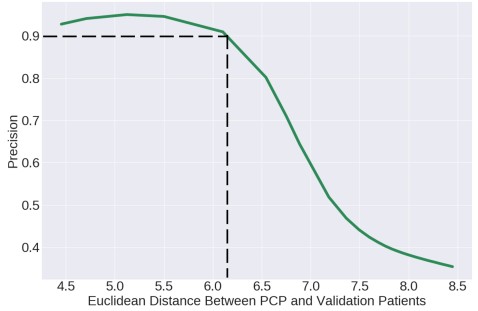

Figure 6: Percentage of pairs of patients, retrieved based on their PCP-derived similarity, with a matching cardiac arrhythmia label. Based on a desired level of precision (e.g., $0.90$), we can identify an appropriate threshold distance between patients (e.g., $d_E \approx 6.2$).

## 5.5 DATASET DISTILLATION VIA PATIENT CARDIAC PROTOTYPES

Our interpretation and the growing evidence we have presented in support of PCPs as efficient descriptors of the cardiac state of a patient led us to investigate the following question: could PCPs be sufficient for training classification tasks, in lieu of the original dataset? This idea is analogous to dataset distillation which focuses on obtaining a coreset of instances that do not compromise the generalization capabilities of a model (Feldman et al., 2018; Wang et al., 2018).

To investigate the role of PCPs as dataset distillers, we train a Support Vector Machine (SVM) on them and evaluate the model on representations of held-out instances. We compare PCPs to three coreset construction methods: 1) **Lucic** (Lucic et al., 2016), 2) **Lightweight** (Bachem et al., 2018), and 3) **Archetypal** (Mair & Brefeld, 2019). In constructing the coreset, these methods generate a categorical proposal distribution over all instances in the dataset before sampling $k$ instances and assigning them weights. For a fair comparison to our method, we chose $k = P$ where $P$ is the number of PCPs. In addition to exploring the effect of these coreset construction methods based on raw instances, we do so based on representations of instances learned via our network. In Table 1, we illustrate the performance of these methods on various datasets.

| Coreset | Chapman | PhysioNet2020 | PTB-XL |
|---|---|---|---|
| *Raw Instances* | | | |
| Lucic | $56.8 \pm 0.8$ | $50.1 \pm 0.1$ | - |
| Lightweight | $56.6 \pm 0.4$ | $50.1 \pm 0.1$ | - |
| Archetypal | $54.8 \pm 0.3$ | $50.1 \pm 0.1$ | - |
| *Representations* | | | |
| Lucic | $57,8 \pm 17.5$ | $50.6 \pm 1.2$ | $51.6 \pm 4.5$ |
| Lightweight | $58.9 \pm 16.8$ | $50.5 \pm 1.2$ | $52.4 \pm 3.6$ |
| Archetypal | $58.1 \pm 16.8$ | $50.5 \pm 1.2$ | $51.0 \pm 5.0$ |
| PCPs | $\mathbf{88.7 \pm 0.5}$ | $\mathbf{52.8 \pm 0.1}$ | $\mathbf{63.5 \pm 0.7}$ |

Table 1: Validation AUC as a function of the coreset strategy. We use an SVM for the Chapman and PTB-XL datasets, and a Random Forest for the PhysioNet2020 dataset given its multi-label classification formulation. In all experiments, the coreset size is equal to the number of PCPs. Results are an average across five seeds.

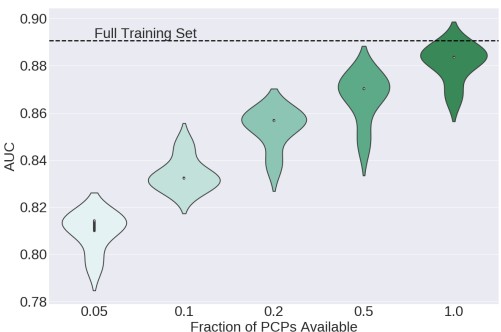

Table 2: Validation AUC after training an SVM on a different fraction of available PCPs ($E = 32$). By training on $100\%$ of PCPs ($N = 6387$), we achieve similar performance to when training on the full training set ($N = 76644$), illustrating the dataset distillation capabilities of PCPs. Results are shown across 5 seeds.

In Table 1, we find that coresets of raw instances generated by traditional coreset construction methods are insufficient for achieving strong generalization performance. For example, the Archetypal method achieves an $AUC = 54.8$ on Chapman. Such poor performance is likely attributed to the poor class separability of the input features. Nonetheless, we show that the exact same set of methods still perform poorly, albeit slightly better, even when constructing coresets from network-derived representations that have been shown to be separable (see Fig. 2a). For example, the Archetypal method now achieves an $AUC = 58.1$ on Chapman. In contrast, we show that PCPs are relatively more effective coresets than those constructed by the remaining methods. On Chapman, for example, PCPs achieve an $AUC = 88.7$. These findings suggest that PCPs have the potential to effectively summarize datasets in a compact manner and act as dataset distillation tools.

Having shown the utility of PCPs as dataset distillers, we wanted to investigate the extent to which further distillation was possible. In Fig. 2, we illustrate the generalization performance of models trained on a different fraction of the available PCPs. For comparison's sake, we also show the AUC of our network when trained on all instances in the training set (**Full Training Set**), which is several folds larger than the number of PCPs. We find that PCPs do indeed act as effective dataset distillers. In Fig. 2, we show that training on $100\%$ of the PCPs ($N = 6,387$) achieves an $AUC \approx 0.89$ which is similar to that achieved when training on the full training set ($N = 76,614$). In other words, we achieve similar generalization performance despite a *12-fold* decrease in the number of training instances. We also show that further reducing the number of PCPs, by selecting a random subset for training, does not significantly hurt performance. For example, training with only $5\%$ of

available PCPs ($N = 319$) achieves an AUC $\approx 0.82$. Concisely, this corresponds to a $7\%$ reduction in performance despite a *240-fold* decrease in the number of training instances relative to that found in the standard training procedure. We arrive at similar conclusions when changing the embedding dimension, $E$ (see Appendix F). Such a finding supports the potential of PCPs at dataset distillers. We hypothesize that this behaviour arises due to our patient-centric contrastive learning approach. By encouraging each PCP to be similar to several representations of instances that belong to the same patient, it is able to capture the most pertinent information and shed that which is least useful.

## 5.6 DISCUSSION AND FUTURE WORK

In this paper, we proposed to learn efficient representations of the cardiac state of a patient, entitled patient cardiac prototypes, using a combination of contrastive and supervised learning. We showed that patient cardiac prototypes are both patient-specific and discriminative for the task at hand. We successfully deployed PCPs for the discovery of similar patients within the same dataset and across different datasets. This opens the door to leveraging clinical information that is available in disparate healthcare institutions. Lastly, we illustrated the potential of PCPs as dataset distillers, where they can be used to train models in lieu of larger datasets without compromising generalization performance. We now elucidate several future avenues worth exploring.

**Obtaining multi-modal summary of cardiac state of patient.** Although our approach was validated on multiple, large, time-series datasets, it was limited to a single modality, the ECG. Incorporating additional modalities to our approach such as coronary angiograms, cardiac MRI, and cardiac CT, may provide a more reliable summary of the cardiac state of the patient. This could ultimately lead to more reliable patient similarity quantification.

**Guiding design of graph neural networks.** Arriving at ground-truth values for the similarity of a pair of patients is non-trivial. Recently, graph neural networks have been relatively successful at discovering and quantifying the similarity of instances, but most necessitate a pre-defined graph structure, which may be difficult to design in the context of physiological signals. We believe that designing this graph structure can be facilitated by our patient-similarity scores, for instance, by using them as an initialization of the edge weights between nodes.

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
