# OpenReview forum: "PCPs: Patient Cardiac Prototypes"
_ICLR.cc/2021/Conference — Reject_

### Official Review · AnonReviewer3 · 2020-10-22
**Several major issues**

**Rating:** 2
**Confidence:** 5

**Review:**

This paper proposes to learn patient-specific representation using patient physiological signals.  The authors design a PCP representation for each patient, which is learned to agree with signals from the same patients and disagrees with the remaining patients. In the supervised part, the classifier is generated from patient-specific parameters by meta-learning. The model was evaluated on three large ECG datasets: PhysioNet 2020 ECG, Chapman ECG, PTB-XL ECG.

Strength:
This is an important problem
The paper is easy to follow, and the experiment settings are well elaborated

___________________________________________________________________________
Weakness:

The authors are ignorant to several lines of existing literature in deep learning for healthcare, particularly deep phenotyping and deep patient subtyping. They also seem not aware of the existing works on explaining Black-box models.   For deep phenotyping algorithms, the patient representations generated from existing deep phenotyping algorithms are ALL patient specific.  To list a few, the authors may take a look at [1-4]. For more related works on deep phenotyping, you may refer to [5]. For more related works on deep learning for ECG data, you may refer to [6]. To learn disease subtypes, patient prototypes, there are also a series of works. Below are several the authors could refer to [7-10]. In addition, there are several model agnostic algorithms designed for explaining deep learning models that could be used here to add more explanability. For example [11-13]. In addition, there are many existing works around contrastive learning on ECG signals, see [14-16]. The authors are either not aware of these lines of works or have wrong understanding on them, which cause several major issues as listed below.

(a) Lack of Novelty. Given the existing works listed below. The novelty of the paper is not enough for the conference. The methodology of the paper consists of a standard contrastive learning method and a standard meta-learning setting. All have been done in one or more existing works.

(b) Lack of Baselines. The paper does not include any baseline. It is unclear how it compares with existing works.

(c) The experiment results are mostly unconvincing. For example,
- In Experiment 5.1, the paper states that “PCPs exhibit tighter clusters than those found with training representations” in Figure 2. However, it is hard to tell which result is more separable from my point of view. A possible suggestion is that the authors could use quantitative clustering metrics to demonstrate, such as Adjusted Rand index. Also, the resolution of figure 2 could be improved.
- In Experiment 5.2,  the detail of meta-learning network, hypernetwork, is not mentioned in the paper. Is it just a matrix transform? Or a more complicated neural network? The detailed implementation of hypernetwork could be discussed.
- In Experiment 5.3, The result is not surprising, since the model have used contrastive loss to maximize similarity of “PCP to Same Training Patient”. Therefore, it should have smaller distances, naturally. Also, we encourage the paper to discuss why using the Euclidean distance not the cosine similarity as mentioned in Sec. 3.1.
- In Experiment 5.4, the paper shows two examples of similar patients, where the performance is hard to evaluate. The reviewer will suggest two ways: (i) using demographics, physiology, or treatment features to match two similar patients, and then quantitatively compare with the matching results given by the paper; (ii) setting distance threshold or cosine similarity threshold to decide whether two patients are similar, and then justify the threshold.
- In Experiment 5.5. This experiment is very problematic. First, the paper claims and show that the PCP representation could provide the same classification performance as training on the whole training set. It is also not surprising, because when obtaining the PCP representations, the model already uses all the training data, then of course PCP would give the same performance; Second, we encourage the paper to compare with using different proportions of the training set. Often the time, the whole training set could be redundant in ECG setting.
- The authors are encouraged to analyze the computational complexity of the proposed method.
- In appendix, figure 9 (bottom) has exactly the same channel signals, which is a significant mistake. Figure 10 has a wrong title description.
- For dataset PhysioNet 2020 ECG, the paper states that “Each recording can be associated with multiple labels”. It is unclear how to use the dataset for evaluation.
- The motivation of the paper could be improved. Also, some related works on contrastive learning are missing, for example [17-18].

References

[1] RETAIN: An Interpretable Predictive Model for Healthcare using Reverse Time Attention Mechanism. NeurIPS 2016

[2] MiME: Multilevel Medical Embedding of Electronic Health Records for Predictive Healthcare. NeurIPS 2017

[3] MINA: Multilevel Knowledge-Guided Attention for Modeling Electrocardiography Signals. IJCAI 2019

[4] RAIM: Recurrent Attentive and Intensive Model of Multimodal Patient Monitoring Data. KDD 2018

[5] Opportunities and challenges in developing deep learning models using electronic health records data: a systematic review, Journal of the American Medical Informatics Association 2019

[6] Opportunities and Challenges in Deep Learning Methods on Electrocardiogram Data: A Systematic Review, Computers in Biology and Medicine 2020

[7] Patient subtyping via time-aware LSTM networks, KDD 2017

[8] DDL: Deep Dictionary Learning for Predictive Phenotyping, IJCAI 2019

[9] PEARL: Prototype Learning via Rule Learning, ACM BCB 2019

[10] Identifying Sepsis Subphenotypes via Time-Aware Multi-Modal Auto-Encoder, KDD 2020

[11] Why should I trust you?: Explaining the predictions of any classifier. KDD 2016

[12] A Unified Approach to Interpreting Model Prediction. NeurIPS 2017

[13] Anchors: High-Precision ModelAgnostic Explanation AAAI 2018

[14] CLOCS: Contrastive learning of cardiac signals. arXiv preprint arXiv:2005.13249, 2020.

[15] A simple framework for contrastive learning of visual representations. arXiv preprint arXiv:2002.05709, 2020.

[16] Subject-aware contrastive learning for biosignals. arXiv preprint arXiv:2007.04871, 2020.

[17] Momentum contrast for unsupervised visual representation learning. CVPR 2020

[18] Representation learning with contrastive predictive coding. arXiv preprint arXiv:1807.03748.

---

> ### Author Response · Authors · 2020-11-18
> **Response to Reviewer 3 - Round 1 (Part 1)**
>
> We thank the reviewer for taking the time and effort to review the manuscript and for providing us with valuable feedback. We address your comments below.
>
> **NOVELTY**
> We would like to respectfully disagree with the reviewer's claim that we lack understanding of the application of deep learning to ECG signals and of papers that are seemingly related to our approach. We describe how our PCPs differ to some of the suggested papers below to ensure to the reviewer that we were aware of such papers and are aware of their distinctions.
>
> We believe the seemingly related work mentioned by the reviewer is actually quite distinct from PCPs. For example, for papers [1-4], these do not exploit contrastive learning, do not learn patient-specific representations (i.e., PCPs) in an end-to-end manner via supervised and contrastive learning, do not leverage patient similarity during inference nor as a tool for retrieving similar (or dissimilar) patients, and do not propose a dataset distillation method. We believe these are substantial and novel contributions that go above and beyond the contributions offered by papers [1-4].
>
> As for SimCLR, CPC, and MoCo, papers [15, 17, 18], these papers are distinct from our work in various ways. They are limited to computer vision, do not learn prototypes in an end-to-end manner, and do not leverage patient information (e.g., patient ID) to learn patient-specific representations. This is just a brief snapshot of how these approaches are different from ours and we would be happy to provide the reviewer with more examples.
>
> As for papers [14,16], although both deal with physiological signals, and learn representations via contrastive learning, they are exclusively focused on pre-training (self-supervised) methods in order to transfer parameters to a downstream task of interest. In contrast, we perform contrastive and supervised learning on the actual downstream task of interest and explicitly learn patient-specific representations in the process.
>
> **BASELINES**
> We modify Sec. 5.5 of the main manuscript to include three coreset construction methods applied to our large-scale ECG datasets and compare them to PCPs. We urge the reviewer to read Sec. 5.5 of the modified version of the manuscript. In a nutshell, we find that PCPs are more effective dataset distillers than the three coreset methods used.
>
> **FIGURE 2**
> We understand how our initial statement of the tightness of the clustering of these representations can be misleading. As a result, we have removed this comment and focus on how PCPs are indeed class-discriminative. Moreover, we replot Fig. 2 with an improved resolution.
>
> **EXPLANATION OF HYPERNETWORK IMPLEMENTATION**
> We dedicate the entirety of paragraph 2 (Sec. 3.2) to discuss how the hypernetwork actually works. Please refer to that section and tell us if anything is unclear. Concisely, in our implementation of this hypernetwork is a single linear layer that maps a representation to a set of parameters. These parameters are then reshaped to match the shape of the parameters in the linear classifier.
>
> **EXPERIMENT 5.3**
> We do not claim that these results are surprising. Nonetheless, we do not think that these results are as expected as you may think. For example, if the contrastive learning procedure did not work, then PCPs would not be closer to representations of instances from the same patient. There might be a high degree of overlap between the 'PCP to Same Training Patient' and 'PCP to Different Training Patient'. The fact that we do not observe this implies that our training procedure was successful in learning patient-specific representations.
>
> In addition to illustrating the Euclidean distance, we also include a figure that illustrates the Cosine distance between representations (Fig. 4). In fact, we show that the separation in the distributions of the distances is even more pronounced when using cosine distance. This is expected since this was the metric that we were optimizing in the contrastive loss.
>
> **PATIENT SIMILARITY QUANTIFICATION**
> We rewrite Sec. 5.4 and include Fig. 6 to better address your concerns. We had originally qualitatively evaluated the ability of PCPs to identify similar patients. To minimize the appearance that such results were 'cherry-picked', we also included results in the Appendix (Appendix D) for three different datasets to support our findings. Moreover, in the modified version of the manuscript, we take the reviewer's advice and quantitatively evaluate the ability of PCPs to identify similar patients. These results are shown in Fig. 6. Concisely, we retrieve patients that are deemed similar based on our PCP approach and determine what percentage of these patients are in fact similar based on their associated cardiac arrhythmia label. We find that, at high similarity values, the precision of our approach is quite high (>90%). Please refer to Sec. 5.4 for more details.
>
> Response continued in Part 2.

---

> > ### Author Response · Authors · 2020-11-18
> > **Response to Reviewer 3 - Round 1 (Part 2)**
> >
> > **EXPERIMENT 5.5**
> > The reviewer claim that such a finding is not surprising. We respectfully disagree with the claim made by the reviewer for the following reason. Just because the PCPs are learned based on the entire dataset, it does not mean that they will effectively capture the pertinent information that is present within that dataset, let alone information that allows it to solve the task at hand and generalize to a held-out set. The fact that training on PCPs (N=6387) performs as well as training on the entire dataset (N=76644) is indicative of its effectiveness in capturing pertinent information. To further support this claim, we refer the reviewer to Table 1 where other coreset construction methods have access to the entire dataset and still perform poorly. Based on your logic, since they are exposed to the entire dataset, they should be effective dataset distillers, which is clearly not the case here.
> >
> > **FIGURE 10 IS NOT A MISTAKE**
> > We would like to clarify that the 12-lead ECG signals presented in (what was Fig. 10) what is now Fig. 9 are not a mistake. This is how we explain this finding. Recall that in all of our experiments, we split patients into training, validation, and test sets using their patient IDs such that they only appear in one of these sets. Therefore, when we exploit our PCPs (which reflect patients in the training set) to identify similar patients in the validation set, we are guaranteed to find a patient (with a different patient ID) that is not present in the training set. In Fig. 9, the retrieved patient in the validation set does indeed have a different patient ID, but upon further inspection we find that this patient’s ECG signals perfectly match those of the patient that is associated with the PCP. The likely explanation is that the organizers of the PhysioNet 2020 dataset erroneously assigned the same ECG signal to two different patient IDs. The interesting part is that our similarity method, driven primarily by the PCPs, was able to discover that error. Although beyond the scope of this paper, this opens the door to potentially applying PCPs for de-duplicating datasets.
> >
> > **CLARIFICATION OF PHYSIONET2020**
> > When we say each recording is associated with multiple labels, we mean that each ECG signal reflects more than one cardiac abnormality and is thus associated with multiple labels. In other words, this is a multi-label classification problem.
> >
> > **MOTIVATION OF PAPER**
> > We rewrite the introduction and methods section of the paper to strengthen the motivation underlying our methods. We urge the reviewer to read the modified version of the manuscript. We also include pertinent contrastive learning literature in our related work, as suggested by the reviewer.
> >
> > Given the substantial changes we have made to the manuscript  based on this reviewer's comments, we urge the reviewer to reread the manuscript and please let us know if anything is unclear.
> >
> > We hope the above responses and the modified version of the manuscript have addressed your concerns.

---

> ### Comment · AnonReviewer3 · 2020-11-21
> **Major issues remain unaddressed. New problem surfaces.**
>
> Thank you for the efforts in addressing some of my minor concerns. However, the most critical issues remain unaddressed. They are listed below.
>
> 1. Lack of novelty and lack of comparison with state-of-the-art methods
>
> The authors claimed “perform contrastive and supervised learning on the actual downstream task of interest and explicitly learn patient-specific representations in the process” is novelty. This is not true.
> -- there are many existing works leverage contrastive learning for ECGs [14][20] and other biosignals [16][19][21-22].
> - combining contrastive and supervised learning: a simple adding up for contrastive and supervised learning in Eq.(5).
> - patient-specific representations are learned in almost all computational phenotyping models I listed in my initial review. They are easily compared and should be compared. Joint self-supervised and supervised methods are also common.
>
> Comprehensive comparison and ablation studies are needed in order to properly evaluate the proposed method. Without that it is unclear how the method performs.
>
> [14] CLOCS: Contrastive learning of cardiac signals. arXiv preprint arXiv:2005.13249, 2020.
> [16] Subject-aware contrastive learning for biosignals. arXiv preprint arXiv:2007.04871, 2020.
> [19] Boosting Generalization in Bio-Signal Classification by Learning the Phase-Amplitude Coupling, CGPR 2020
> [20] Learning Generalizable Physiological Representations from Large-scale Wearable Data, Machine Learning for Mobile Health workshop at NeurIPS 2020.
> [21] Uncovering the structure of clinical EEG signals with self-supervised learning
> [22] Self-supervised representation learning from electroencephalography signals
>
>
> 2. Problematic results
>
> Based on cosine distance in new Figure 4, PCPs retrieval/matching in inference is unfortunately very problematic. We can observe that the cosine distance to Validation patients (purple distribution) is centered at 1.0 with very narrow band, which means the learned “old” PCP representations are perpendicular to “new” patient representations almost surely (which makes sense in high-dimension statistics). During inference, as far as the reviewer understand, the hypernetwork meta-parameters are retrieved by finding the best PCP match to the current patient representation. If the learned PCPs are perpendicular to new patient representations (meaning that they are irrelevant), then the question is: why is the matching useful?

---

### Official Review · AnonReviewer4 · 2020-10-28
**Intuitive, however, clinical utility maybe limited as presented**

**Rating:** 7
**Confidence:** 3

**Review:**

Intuitive concept and interesting study.

Some general comments.

It would be helpful to know why authors used cosine similarity score as opposed to other metric. Just because of its simplicity? explain.

What is the difference between this work and a clustering approach, this seems to be somehow a clustering solution they are providing. I suggest having a comparison with existing phenotypic (clustering) that have been proposed and show how this new model is better (if) and if such comparison is unreasonable, then I except to have a clear justification on why and how in theory this is different and perhaps better.

The motivation can be further emphasized. I understand the rational given my experience working in healthcare setting; however, other readers may not see the true value of this project as presented.

The English is at time very assertive and strong use of statement, where instead the patten of distributions are not confirming but rather corroborating since the experiments are not systemic but rather authors show few distributions that seem to demonstrate what they want to highlight. I do not say, the authors are cherry picking but the way it is presented can be misleading.

In terms of manuscript, the abstract is too short and should have the typical sections for the reader to get a sense before jumping to the full manuscript. Furthermore, as abstracts are usually used in automated searches and meta-analysis, is highly important to have a clear and comprehensive abstract (background, method, results, discussion and conclusion).

Code should be made available for this submission.

---

> ### Author Response · Authors · 2020-11-18
> **Response to Reviewer 4 - Round 1**
>
> We thank the reviewer for taking the time and effort to review the manuscript and for providing us with valuable feedback. We address your comments below.
>
> **COSINE SIMILARITY SCORE**
> We leverage cosine similarity in two parts of the manuscript. The first involves using the cosine similarity to quantify the similarity of instance representations to PCPs (Eq. 1). Our motivation for using this metric stems from its utility which has been empirically shown in the contrastive learning literature (e.g., https://arxiv.org/abs/2002.05709). Our second use of the cosine similarity is during the inference stage (Eq. 4) when we calculate the similarity of validation instance representations to PCPs that have already been learned. In this setting, we remain consistent with the similarity metric that is used during training in order to make sure that the similarity values are on the same scale as those obtained during training.
>
> **PCPs ARE DISTINCT FROM CLUSTERING**
> We clarify how PCPs are distinct from clustering in the modified version of the manuscript (Sec. 3.2, second paragraph). It is worthwhile to note that although this approach bears some resemblance to clustering, it is distinct from it. In a clustering scenario, we would have assigned labels to instances based on their proximity to PCPs. In contrast, we are leveraging this proximity to determine the input of a hypernetwork (see Sec. 3.2, Fig. 1 right).
>
> **HEALTHCARE MOTIVATION**
> We modify the introduction slightly to strengthen the motivation behind our proposed methods. We urge the reviewer to read the modified version of the manuscript and please let us know if the motivation would still be considered unclear for a machine learning practitioner.
>
> **ABSTRACT**
> We take the reviewer's advice and lengthen the abstract to contain important components of the manuscript.
>
> **CODE**
> We will make this code available.
>
> We hope the above responses and the modified version of the manuscript have addressed your concerns.

---

### Official Review · AnonReviewer1 · 2020-10-31
**Important application domain but unclear contribution**

**Rating:** 5
**Confidence:** 4

**Review:**

ICLR PAPER

PCPs: Patient Cardiac Prototypes


##########################################################################

Summary:


The paper proposes unsupervised neural network models to learn patient-specific representations for ECG applications. The architecture implements ideas from prototype networks and contrastive learning in order to discover similar patients across datasets and compress datasets based on the representations only.

##########################################################################

Reasons for score:


Overall, I vote for marginal reject. The paper is very well written and I like the idea of leveraging the representations of ECG data but my major concerns are the difference in terms of novelty when compared to other concurrent papers submitted to this venue, namely DROPS, CLOPS (arxiv, Kiyasseh et al.), and CLOCS (arxiv, Kiyasseh et al.). They all seem to use the same datasets and building upon the contrastive learning paradigm. The tasks might be slightly different however all share significant common ground.


##########################################################################Pros:


1. The paper leverages one of the most important and overlooked data modalities, that of ECG. The community should invest more in models tailored to this kind of sensor data.

2. This paper provides comprehensive experiments, including both qualitative analysis and quantitative results, to show the effectiveness of the proposed framework. I especially enjoyed the euclidean distance plots.


##########################################################################

Cons:


1. My major concern, as described above is the novelty compared to other methods proposed recently, possibly by the same team. A comprehensive comparison/ablation across all these methods is needed in order to evaluate what achieves better results in which task.

2. Also, I struggle to understand if the resulting PCPs are fewer data points than the original training set. As far I understand, a patient generates a very long time series of ECG which has to be segmented to chunks or windows. Now, PCPs compress all these windows to a single latent vector (?). In this regard, why do we need the fancy hypernetwork? Could we just average all latent representations of the same patient after training is done? It seems that the paper is using a "mean" baseline but this is still part of the PCPs, not all training samples.

3. In terms of tasks, the paper presents the patient similarity and the dataset distillation as the most important applications of this method. However there are no comparisons with any other methods, for example, established methods from distillation (teacher-student, core-sets).


##########################################################################

Questions during the rebuttal period:


Please acknowledge all concurrent papers using similar datasets/methods/tasks.

The hypernetwork module is not motivated sufficiently.

Are there other tasks that this method could be useful to?

---

> ### Author Response · Authors · 2020-11-18
> **Response to Reviewer 1 - Round 1**
>
> We thank the reviewer for taking the time and effort to review the manuscript and for providing us with valuable feedback. We address your comments below.
>
> **NOVELTY**
> We would like to clarify that PCPs are distinct from seemingly related contrastive learning methods that have been introduced recently.
>
> CLOCS implements contrastive learning (purely self-supervised) as a pre-training method on large-scale ECG datasets with the aim of transferring parameters to a downstream task of interest. In contrast, we leverage contrastive and supervised learning simultaneously to solve the downstream task of interest. PCPs is not a pre-training method. Moreover, contrastive learning methods like CLOCS depend on a certain number of negative samples when calculating the noise-contrastive estimation loss, a feature that has been shown to limit performance (e.g., with SimCLR and MoCo papers). Instead, in PCPs the number of negatives is fixed based on the number of prototypes one learns from the dataset. For example, if there are P prototypes, then there will be P-1 negatives for each representation. Lastly, PCPs are efficient descriptors of the cardiac state of the patient and can be leveraged for dataset distillation. In contrast, CLOCS lacks that ability.
>
> Based on our reading of DROPS, it is also distinct from PCPs. In DROPS the authors learn attribute-specific prototypes based on ECG time-series data. These attributes are based on cardiac disease, sex, and age. In contrast, with PCPs we learn representations that do not explicitly account for attributes and are instead patient-specific. Moreover, these patient-specific representations are leveraged for patient similarity quantification and dataset distillation, two actions that DROPS does not perform and cannot do so at the same level of granularity (i.e., at the patient level).
>
> CLOPS is a continual learning method, and not a contrastive learning method. These approaches are only similar in that they use the same datasets and pre-processing steps. We follow their approach when it comes to dataset usage and pre-processing to ensure reproducibility in the field of clinical machine learning. Beyond that, CLOPS looks to avoid catastrophic forgetting in the context of learning from data that stream over time. With PCPs, data are available all at once, and we are learning end-to-end descriptors of the cardiac state of the patient.
>
> **CLARIFICATION REGARDING PCPs**
> We have modified Sec. 3.1 and 3.2 to improve clarity and address your concerns. We urge the reviewer to read those sections in the modified version of the manuscript.
>
> To clarify any potential confusion surrounding the number of PCPs that are learned in an end-to-end manner, we present the following: each patient generates a 12-lead ECG signal. This signal, as the reviewer had mentioned, is split (a) temporally by creating temporal chunks/windows, and (b) spatially by separating the 12 leads. Therefore, if each 12-lead ECG signal were split into 2 temporal chunks and 12 leads, then each patient would have a total of 2*12 = 24 ECG segments. PCPs attempt to compress all of these 24 ECG segments to a single representation for each patient. Fig. 2 illustrates this compression qualitatively as you transition from a crowded t-SNE to one with fewer points.
>
> **REASON FOR HYPERNETWORK**
> One of the driving forces behind the paper was the desire to generate patient-specific predictions. The importance of patient-specific predictions is outlined in the introduction wherein we discuss the detriments of population-based findings that are all too common in the medical domain. To obtain such patient-specific predictions, we aim to generate patient-specific parameters. We opted to use a hypernetwork since it is a module that broadly generates parameters. To make those parameters patient-specific, however, we conditioned the hypernetwork on PCP(s). Since PCPs were learned in such a way, and shown, to be patient-specific, this justified our use of PCPs as inputs to the hypernetwork.
>
> **BASELINES FOR DATASET DISTILLATION EXPERIMENT**
> We modify Sec. 5.5 to incorporate 3 baseline coreset construction methods from the literature and compare it to the performance of PCPs. We urge the reviewer to read Sec. 5.5 and in particular to view the results in Table 1. In a nutshell, we illustrate that PCPs outperform the coreset construction methods on the three large-scale ECG datasets used in the paper.
>
> We hope the above responses and the modified version of the manuscript have addressed your concerns.

---

### Decision · Program_Chairs · 2021-01-07
**Final Decision**

**Decision:**

Reject

**Comment:**

The paper proposes the use of contrastive learning to learn patient specific representations from medical data. The authors show how their method can be used to find similar patients within and across datasets.

The paper has some issues, as indicated by the reviewers:
- similarity to past work; in the response to R1, the authors specify differences to related papers; however, experimental comparisons should still be performed against CLOCS and DROPS
- the evaluation is not fully convincing (the follow up comments of Reviewer 3), including the retrieval of similar patients